# Task Planner for Robotic Disassembly of Electric Vehicle Battery Pack

**Martin Choux** *,†, **Eduard Marti Bigorra** †,‡ **and Ilya Tyapin**

Department of Engineering Sciences, University of Agder, 4604 Kristiansand, Norway;
edumabi@gmail.com (E.M.B.); Ilya.tyapin@uia.no (I.T.)
* Correspondence: martin.choux@uia.no
† These authors contributed equally to this work.
‡ Current address: Northvolt AB, 112 47 Stockholm, Sweden.

**Abstract:** The rapidly growing deployment of Electric Vehicles (EV) put strong demands on the development of Lithium-Ion Batteries (LIBs) but also into its dismantling process, a necessary step for circular economy. The aim of this study is therefore to develop an autonomous task planner for the dismantling of EV Lithium-Ion Battery pack to a module level through the design and implementation of a computer vision system. This research contributes to moving closer towards fully automated EV battery robotic dismantling, an inevitable step for a sustainable world transition to an electric economy. For the proposed task planner the main functions consist in identifying LIB components and their locations, in creating a feasible dismantling plan, and lastly in moving the robot to the detected dismantling positions. Results show that the proposed method has measurement errors lower than 5 mm. In addition, the system is able to perform all the steps in the order and with a total average time of 34 s. The computer vision, robotics and battery disassembly have been successfully unified, resulting in a designed and tested task planner well suited for product with large variations and uncertainties.

**Keywords:** robotic disassembly; electric vehicle battery; task planner

## 1. Introduction

As the adoption rate for electric vehicles (EV) is now accelerating worldwide, EV Lithium-Ion Batteries (EVBs) repurposing or recycling volumes are expected to be larger in 5–10 years (120 GWh/year available by 2030 [1]) and legislation will likely demand higher collection and recycling rates as for example in the newly proposed regulation of the European parliament and of the council concerning batteries and waste in December 2020. Today, the automotive Lithium-Ion Batteries (LIBs) dismantling process is mainly carried out manually and the use of robotics in this process is limited to simple tasks or human assistance [2]. These manual processes are time consuming and must be done by highly skilled personnel. As a direct consequence, the manual total disassembly of Li-ion EVBs might not be profitable and would be stopped at an optimal level, i.e., partial disassembly, that achieves maximum profit while decreasing the environmental impact [3]. In comparison, automated systems are more robust, have a lower-cost, reduce injuries and/or sickness, make the workplace more attractive for those hard-to-recruit-and-retain skilled workers, and are best suited for up-scaling to high-volumes. Therefore, fully automated disassembly of EVBs is inevitable. The main challenges for the success of the automated systems in dismantling are the variations and uncertainties in used products [4]. These challenges in the robotic disassembly of Electrical and Electronic components in electrical vehicles have been presented in article [5] where the need for cognitive systems is identified to enhance the effectiveness of automated disassembly operations. In the case of automated disassembly of EV batteries, advances in Computer Vision (CV) and cognitive robotics offer promising tools but this topic remains an open research challenge [6]. The

disassemblability of industrial batteries, as described in articles [7,8], can be improved either by modifying their design and increasing its standardisation, for example redesigning a battery module to make it remanufacturable [9], or by developing new technologies to ease and eventually remove some of the challenges, as for example, making the recognition of fasteners an easy task. This second route, i.e., making the disassembly process smarter and more efficient through better cognitive capabilities is the one chosen in this research, motivated by the fact that many EV battery innovations are emerging making the design of modules and packs prone to rapid changes.

Over the last 15 years, research has been conducted on the recycling of Lithium-Ion Batteries (LIB) cells, mostly focusing on the mechanical and metallurgical recycling processes [10]. However none of the described recycling methods is integrating robotic disassembly in their pre-processing of EVBs, i.e., processes which does not alter the structure of the LIB cells, and the mechanical or pyrometallurgical processes start with EVB cell modules as input. However, a large portion of metals to be remanufactured or recycled comes from housings (pack and module), electrical wire and connectors. For a Nissan Leaf first generation, the weight of the cells alone represents only 60% of weight of the total battery pack [6]. The disassembly process has been extensively studied in the literature, as shown in the survey [11] where disassembly processes of Waste of Electrical and Electronic Equipment (WEEE) are also present [5,12]. In article [13] authors presented an automatic mechanical separation methodology for End-of-Life (EOL) pouch LIBs with Z-folded electrode-separator compounds (ESC). Customised handling tools were designed, manufactured, and assembled into an automatic disassembly system prototype. While this aspect is still an active research field, the focus is now shifting towards automated solutions to support the whole recycling chain. Industrial solutions for the automated disassembly of battery-operated devices have been implemented, but they are limited to specific and often small-sized products. Apple has implemented an automated disassembly line for Iphone6 [14], however the process is not flexible nor adaptive and can only disassemble one model phone in perfect conditions. In article [15], cooperative control techniques are developed and demonstrated on the robotic disassembly of PC. In article [16], a vision system to identify components for extraction and simple robotic processes are used to disassemble printed circuit boards. Using visual information to automate the disassembly process is further developed with the concept of cognitive robotic systems [17] and is applied for disassembly of Liquid Crystal Display (LCD) screens [18].

Building upon the existing work in this area, this paper aims to improve the design of the task planner responsible for automatically generating the disassembly plan and sequences without precise a priori knowledge of the product to dismantle. The developed functions are presented and the results are validated through experiments conducted on a Hybrid Audi battery pack.

## 2. Materials and Methods

### 2.1. Task Planner Design

The experimental setup is composed of IRB4400 robot (ABB, Zürich, Switzerland), IRBT4004 track (ABB, Zürich, Switzerland), and Zivid One 3D camera (Zivid, Oslo, Norway) mounted on the robot arm, all connected to a PC with Ubuntu and running the Robot Operating System (ROS), and a A3 Sportback e-tron hybrid Li-ion battery pack (Audi, Ingolstadt, Germany). More information about the connection setup and use of the ROS as a middleware can be found in article [19], whereas a complete description of the Audi battery pack and its disassembly sequences has been presented in article [3]. The task planner proposed in this paper and shown in Figure 1 analyses the information provided by the vision system based on Zivid camera, makes decisions regarding dismantling actions and sends a predefined path to the industrial robot's controller.

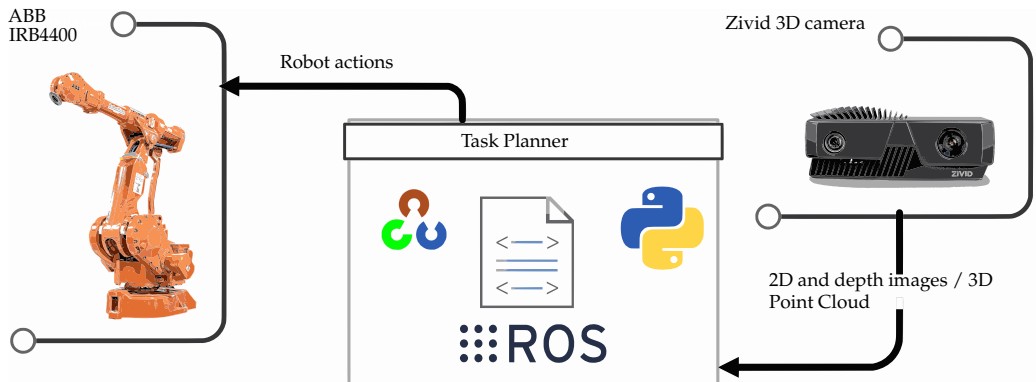

**Figure 1.** Task planner concept.

The main loop of the task planner is organised as the following: takes 2D and 3D images, detects and identifies components, finds the component's positions in the world reference frame, defines an order of operations, removes the components, and repeats this actions until the goal state is reached (refer to steps A–E respectively in Figure 2).

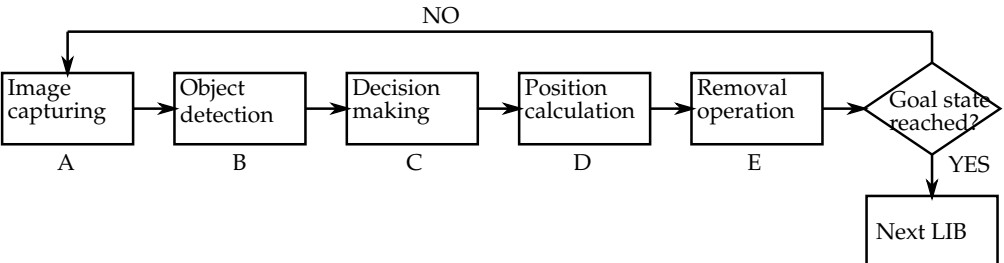

**Figure 2.** Task planner main loop.

### 2.1.1. Image Capturing (A)

The task planner moves the robot into several predefined poses to ensure that the system is able to observe different parts of the LIB pack. In order to reach better accuracy, up to eight pictures are required, especially when screws placed in the lateral sides of the battery are present.

### 2.1.2. Object Detection (2D Image) (B)

Different components in the image such as screws, battery modules, connecting plates, and Battery Management System (BMS) are detected using the YOLO (You Only Look Once) algorithm [20] which also provides the bounding box positions of all constituting components. The output of the object detection procedure as shown in Figure 3 is a file containing all detected objects, labels and corresponding coordinates.

The positions of the detected objects in different images are merged using a weighted mean of each positions.

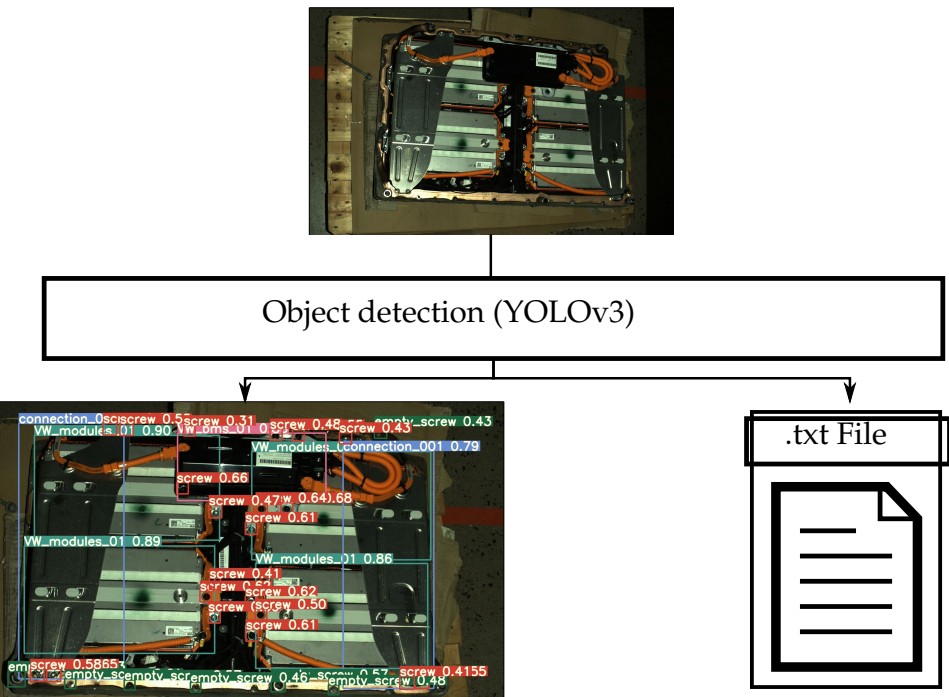

**Figure 3.** The YOLOv3 algorithm takes as input the 2D image of the EV Lithium-Ion Battery (EVB) pack, detects the components, finds the bounding box coordinates and class probabilities, and store the information in a text file. The labels shown on the picture have one color for each class.

### 2.1.3. Decision Making (C)

In order to set a sequence of the removal operations, only one image is used, which is taken with lower camera angle with respect to a horizontal plane. The images with a higher inclination are used for object detection only. A list of the component positions in the removal order is created by adding the detected screw positions and following by a computer vision analysis of each specific component. Based on the probability of being over the other components the remaining parts are added in the list in a correct disassembly order.

### 2.1.4. Position Calculation (D)

At this point the system defines the positions of the objects in 2D camera coordinates (pixels). In order to move the robot to those positions, they must be converted into 3D points. The 2D object coordinates in camera frame are transformed into 3D coordinates using the depth information of the 3D vision system. Then, the captured positions are known, so that the world reference position of the objects can be obtained. This action is done for all images. Once all the positions from the different points of view are found the nearby points (representing the same object or component) are merged.

### 2.1.5. Robot Communication and Removal Operation (E)

Once the order of the operations and all the positions are known, the last step is to be proceed in dismantling. The task planner calls the removal operation of every single component. Details on the design of the removal operation are not presented in this paper, since a scope is limited to moving the robot arm onto the calculated positions, where the selected removal operation based on the classification of the component is achieved.

### 2.2. *Main Functions*

The main functions and scripts used by the task planner connect the computed information by the mean of tests and loops as shown in the complete flow chart in Figure 4.

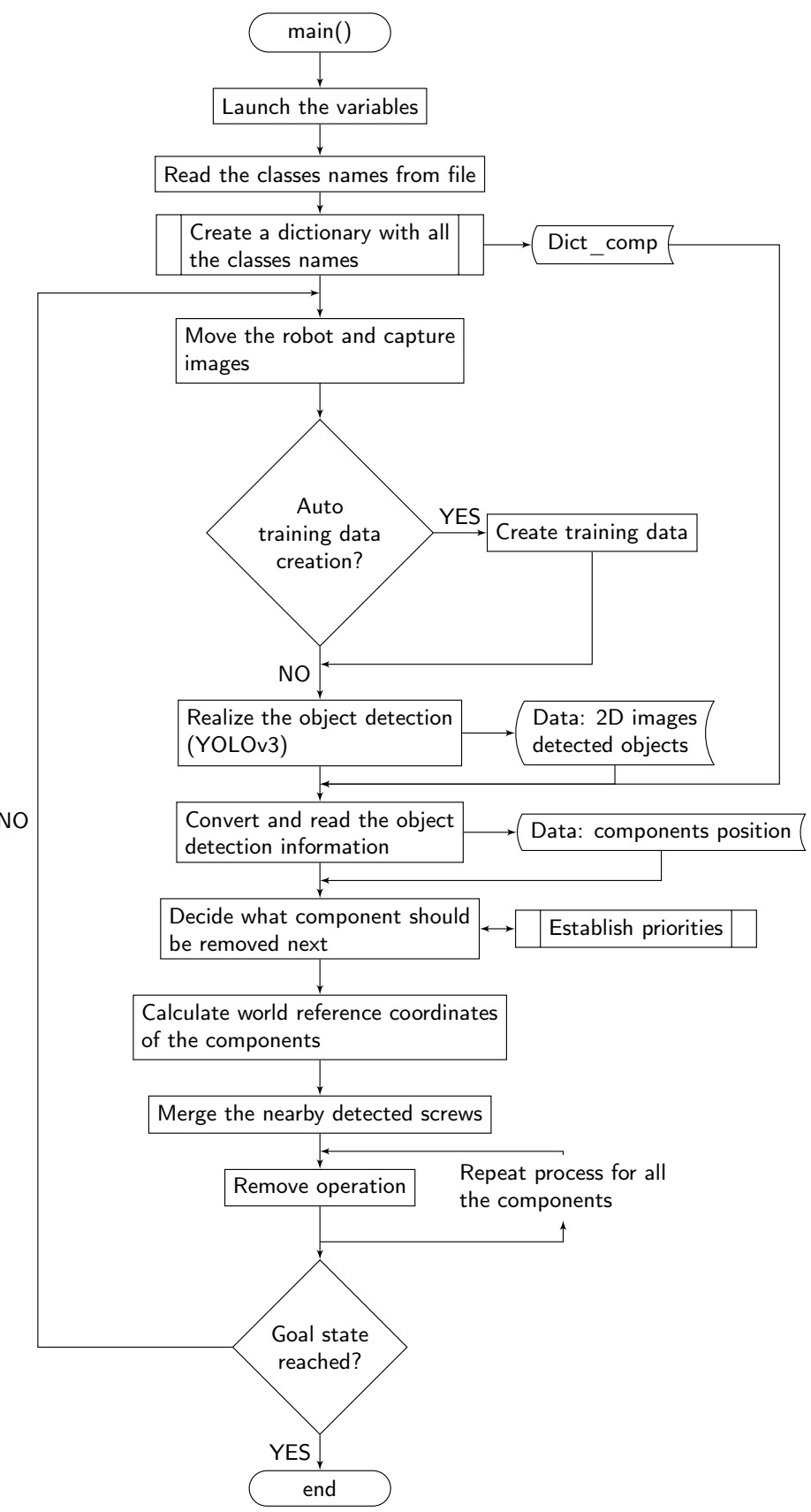

**Figure 4.** Task planner flow chart.

### 2.2.1. Function: *main()*

Fist of all, the main function *main()* whose flow chart is shown in Figure 4, declares and initialises all the variables used and transferred in and between the subsequent functions, as for example the screws, connecting plate, Battery Management System (BMS), or module positions and number. Then, it reads the file containing the classes names (classes.names), and saves them into a dictionary *(dict_comp)*. The creation of this dictionary aims to allow the system to have access to the nomenclature in order to relate the detected classes with their name and characteristics necessary for further analysis.

Once the dictionary is created, the function starts its principal loop. Note that this loop keeps running until the dismantling is completed. The first action of the loop is to move the robot to predefined positions and take 3D images, i.e., XYZ + color (RGB) + quality (Q) for each pixel, of the battery pack. Next, once the algorithm has been trained, it runs the object detection to detect the components placed in these images, using the YOLOv3 algorithm.

After the detection, classification and pose estimation of the objects in the image frame are done. The function analyses each taken image, finds different components and their characteristics from Dict_comp, and converts the positions of the objects from the image base frame to the camera reference frame. Following this step, a sub-process establishes priorities to enable the decision-making operation on what component should be removed next. The respective functions are named *what_component* and *has_comp_over* and are further described in the next section. Thus, when the detected positions have been converted into the camera frame, they are further transformed into the world reference frame coordinates. At this stage, since several 3D pictures of the same components have been captured, the resulting multiple positions of the same components are merged, which increases the position accuracy. The output of this function is the positions of the different components in world reference frame. The components are then placed in order of removal preference and finally, the task planner's main function runs the removal operations for all the detected component before starting the main loop again if the goal state is not reached.

### 2.2.2. Cognitive Functions: *what_component()* and *has_comp_over()*

The function *what_component()* flow chart is shown in Figure 5 and the function is responsible for the decision making. The function analyses the detected objects, and decides the order of the removal operations using the computer vision based sub-function *has_comp_over()* . The sub-function *has_comp_over()* flow chart is shown in Figure 6, and the function establishes different probabilities of a specific component to have a component over.

The inputs for the function *what_component()* are:

- *screw_pos* (array): An array containing coordinates of detected screws, referred to the image base frame.
- *connect_pos* (array): An array containing coordinates of detected connective components, referred to the image base frame.
- *compon_pos* (array): An array containing coordinates of detected general components, referred to the image base frame.
- *empty_screw_pos* (array): An array containing coordinates of detected empty screw holes, referred to the image base frame.
- *co_tot* (array): An array containing coordinates of all detected components, referred to the image base frame.

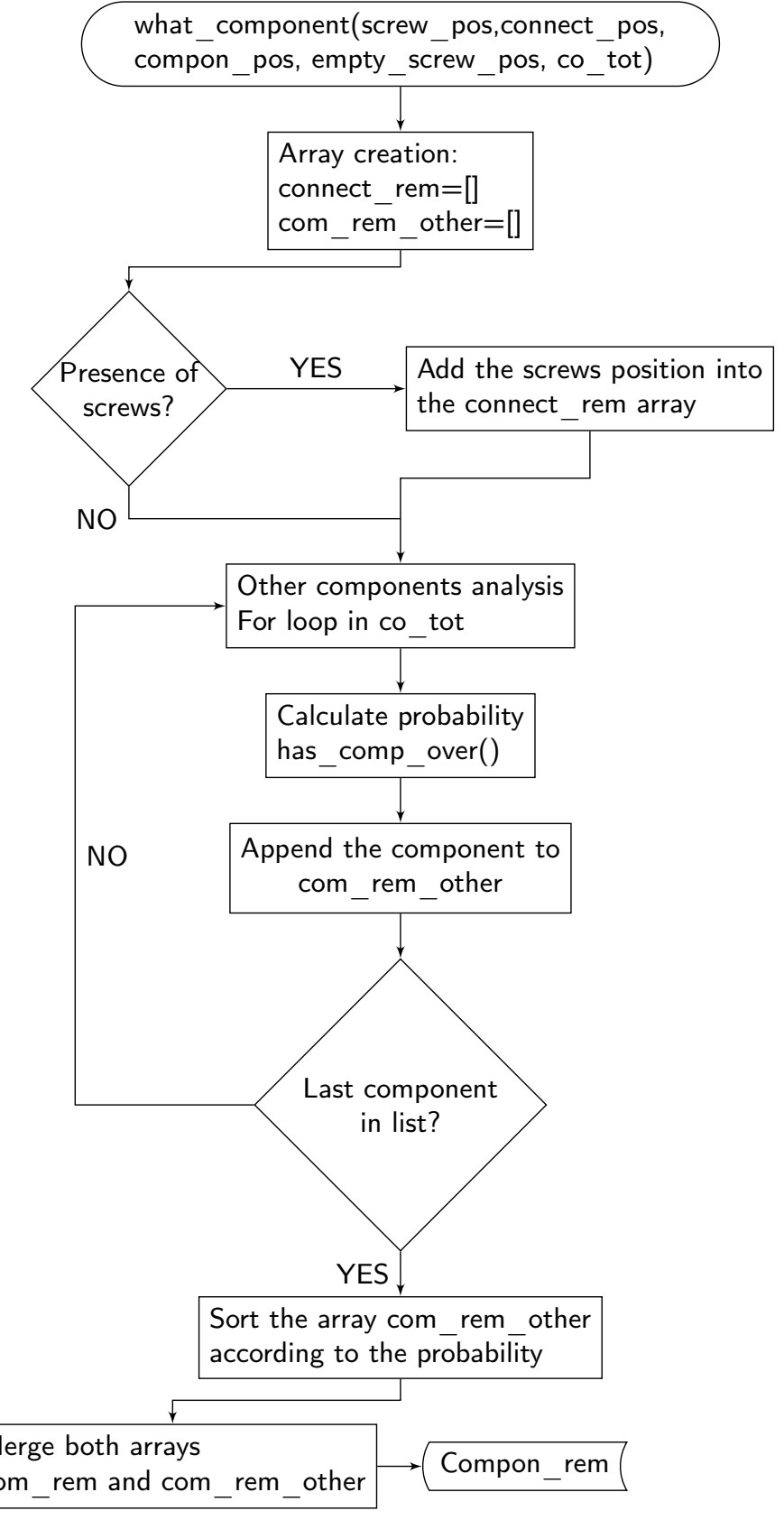

**Figure 5.** Function *what_component* flow chart.

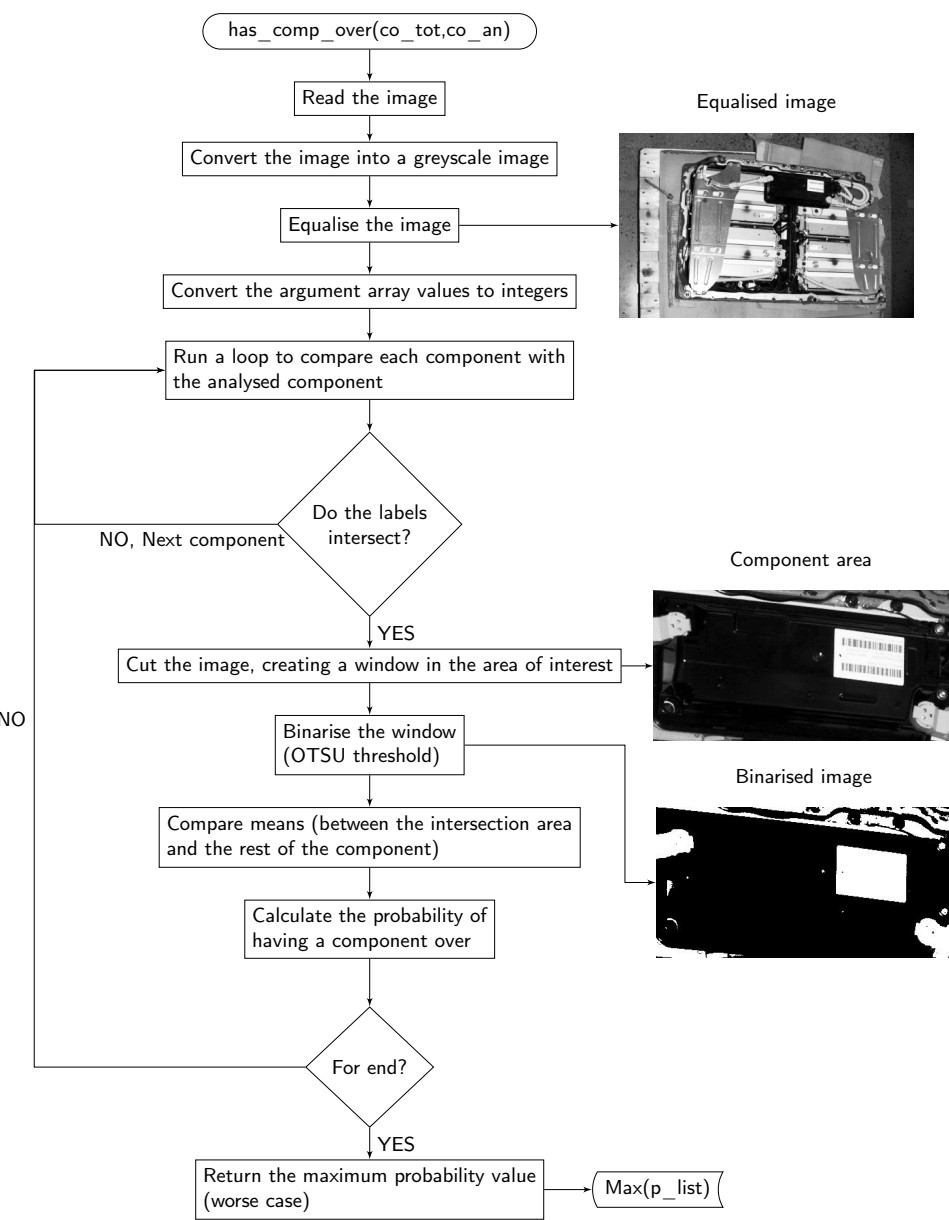

**Figure 6.** Function *has_comp_over* flow chart.

First of all, the function creates the variable *compon_rem* (array). The aim of this array is to contain the list of the components positions in the removal order. Given the nature of the disassembly process, the first components to be removed are screws, thus, their positions are the first to be added to the *compon_rem* array.

To add the rest of the components and decide the order of the operations, the function *what_component* runs over the list co_tot analysing each component using the function *has_comp_over*.

Essentially, *has_comp_over* realize a computer vision analysis of each specific component, and returns the probability of being over the other components. Thus, the components are ordered from more probability to be over to less probability.

The inputs for the sub-function *has_comp_over* are:

- *co_tot* (array)
- *co_an* (array): Array containing the image coordinates of a specific component (the component being analysed).

The function *has_comp_over* runs a nested for loop over the list containing all the components, analyzing the ones that are intersecting the component that is being inspected in the outer for loop.

To realise this comparison, the full image is converted into grey-scale and equalised. The output image contains a better distribution of the intensities maintaining the relevant image information [21].

After obtaining the equalised image, the function crops the image into the area of interest, in this case, the area of the analysed component. Then, segmentation is applied to the window, binarising it using an Otsu threshold. In the Otsu method, the threshold is determined by minimizing intra-class intensity variance [22].

When the window has been binarized, the mean of the pixel intensities in the intersection area is compared to the means of the intersecting components areas excluding the intersection. The component for which these two means (intersection alone and component area excluding intersection) are the most similar is the most probable to be on the top layer, i.e., over all the others, and hence to be removed first. An example with two overlapping components is shown in Figure 7. In this example, component 2 is over component 1 because:

$$||f(I) - f(A_2 - I)|| < ||f(I) - f(A_1 - I)|| \tag{1}$$

where $f(A)$ is a function calculating the mean of pixel intensity in area A.

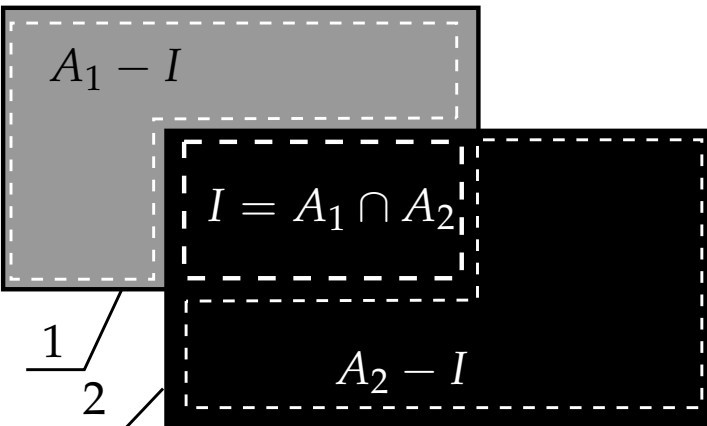

**Figure 7.** Component 2 with total area $A_2$ is overlapping component 1 with total area $A_1$. The intersection area is $I$.

After repeating this procedure for all the components, the function returns the maximum value, max(p_list), of the calculated probabilities of having another component overlapping the analyzed one.

### 2.2.3. Merging the Pictures: *merge_detection()*

The function *merge_detection* aims to merge positions of the components (for this version only the screws) commonly detected in one or more images. The function flow chart is shown in Figure 8.

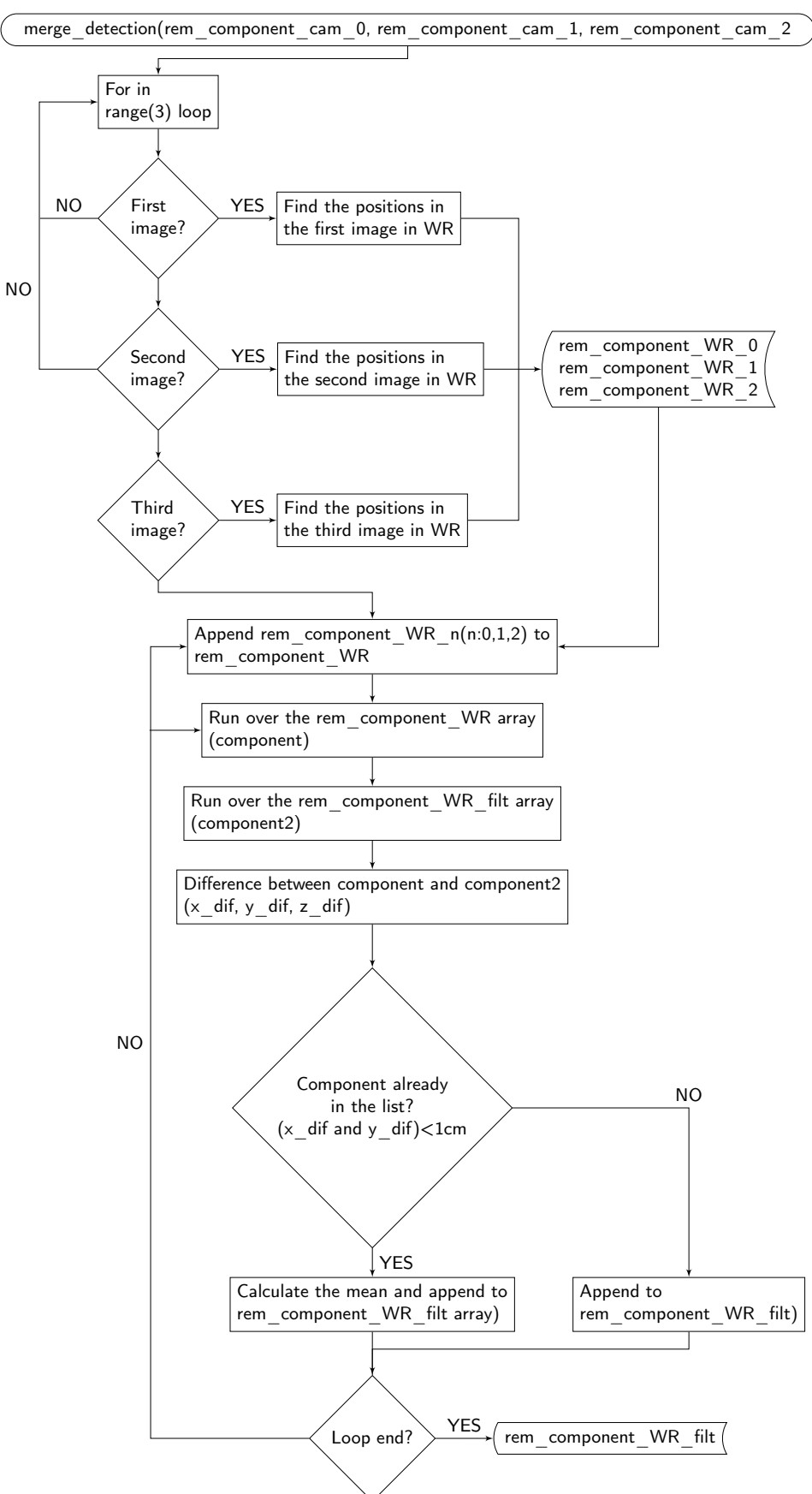

**Figure 8.** Function *merge_detection flow chart.*

The inputs of the function *merge_detection* are:

- *rem_component_cam_0* (Array): Array containing positions of all screws detected in the first image in camera reference frame.
- *rem_component_cam_1* (Array): Array containing positions of all components detected in the second image in camera reference frame. The components are placed in the removal order list, because the decision-making has been previously done in this image (given its inclinations and consequently its light conditions).
- *rem_component_cam_2* (Array): Array containing positions of all screws detected in the last image in camera reference frame.

The *merge_detection()* function runs a "for in range" loop to find positions of components in the world reference frame. It calls the *WR_pos* function, where the positions are stored in the variable *rem_component_WR* including repeated positions.

In the next step the function merges the repeated positions to find the output list (*rem_component_WR_filt*). In order to filter the points, the function runs over the *rem_component_WR* array adding not-repeated components to the filtered list. The function considers two components as one, when the x and y distances are lower than 1cm, and defines the final position as the mean of both points.

## 3. Results

The aim of this section is to validate the order suggested by the task planner and to characterise its performance regarding time and accuracy. The Audi A3 Sportback e-tron Hybrid Li-ion Battery Pack serves as the case study. The description of the EVB pack and its components as well as the disassembly process of the battery are detailed in article [3] whereas Table 1 presents the composition, i.e., relative weight of each components and materials. For safety reason when testing the concept, all modules have been manually discharged separately. However, when integrated in the pilot plant, the EVB packs will be discharged prior complete disassembly at a certain state of charge depending if the modules are to be repaired, re-purposed, re-manufactured, or recycled. In addition, damaged EVBs that represent high risk for thermal runaway or gas emission will be sorted out. These last two steps are outside the scope of the present work.

**Table 1.** Audi A3 Sportback e-tron battery pack constructive components and materials.

| Component | %*w/w* | Material |
|---|---|---|
| Upper housing shell | 2.8% | Composite |
| Upper and lower insulator | 0.4% | Expanded polyethylene |
| BMS (Battery Junction Box and Battery Management Controller) | 2.2% | Plastic, electronics |
| Connecting plates (Top transverse covers) (2) | 0.7% | Al |
| Modules (8) | 75.3% | Li, Co, Mg, Ni, Cu, Al, Graphite, plastic |
| Cooling system | 0.4% | Ruber, plastic, Al |
| Lower housing shell | 16.6% | Al |
| High-voltage cables and connectors | 1.2% | Cu |
| Screws | 0.4% | Fe |

### 3.1. Object Detection Results

Figure 9 shows the results of the YOLOv3 algorithm implementation, where the red filled boxes indicate the position of the connective components, the blue-filled boxes indicate the screws positions, the pink-filled boxes indicate the position of the BMS and the black-filled boxes indicate the position of the battery modules.

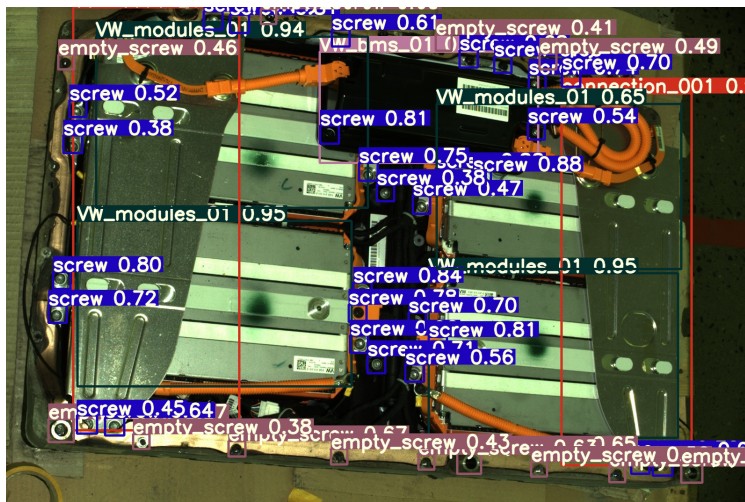

**Figure 9.** YOLOv3 output results: all screws, modules, transverse covers, and BMS are detected, classified, and localised.

### 3.2. Time Analysis

The timings of the operations realized by the task planner, i.e., image capture, object detection, decision making, and motion to estimated pose, have been recorded on 20 repetitions with the physical setup, resulting in the mean times summarized in Figure 10. During experiments, the speed of the industrial robot has been reduced to 25% speed for safety reasons. The expected timing at production speed (100% speed) are shown in black and red in Figure 10.

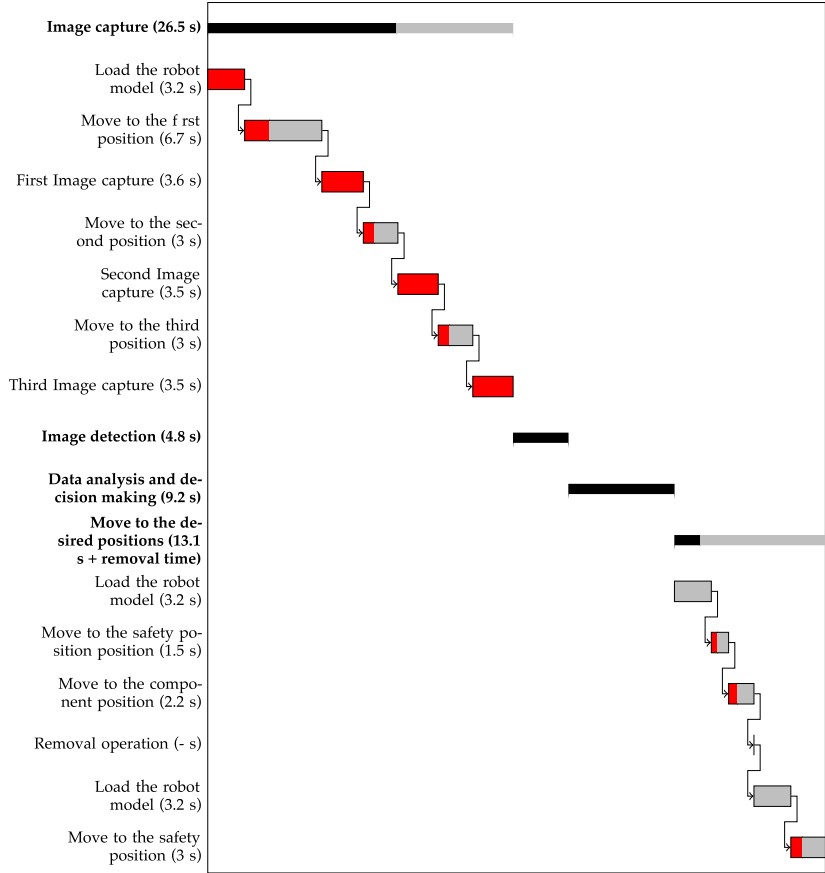

**Figure 10.** Timing summary at 25% speed in grey and expected timing at full speed in black and red.

### 3.2.1. Image Capture (Mean Time: 29.1 s)

In this case, the image capture process refers to the robot movement into the image taking positions and image capturing. In order to implement the capturing, the process has been divided into seven different actions. The first action refers to the robot model loading, and the rest refer to the robot movements and image captures, see Figure 11.

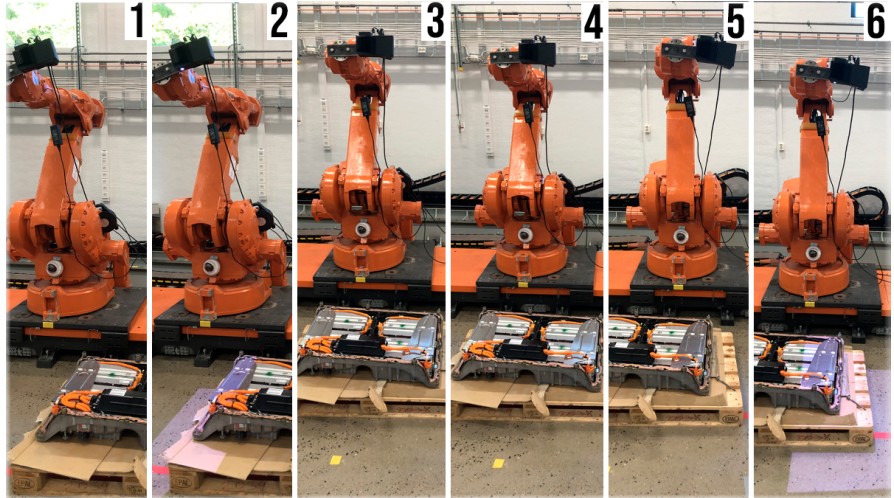

**Figure 11.** Image capturing actions.

1.  Move to the first position. Mean time: 6.7 s.
2.  First image capture. Mean time: 3.6 s.
3.  Move to the second position. Mean time: 3.0 s.
4.  Second image capture. Mean time: 3.5 s.
5.  Move to the third position. Mean time: 6.0 s.
6.  Third image capture. Mean time: 3.6 s.

### 3.2.2. Object Identification (Mean Time: 4.8 s)

In this stage the YOLO algorithm is applied to three taken images to detect and identify different components on 2D images. Mean time: 4.8 s. In this study 3 images are analysed but up to 8 images are required to increase an accuracy.

### 3.2.3. Data Analysis and Decision Making (Mean Time 9.2 s)

Data analysis and decision-making refers to calculating the object positions in the world coordinate frame and define the optimal path for the operations. Mean time: 9.2 s.

### 3.2.4. Move to the Desired Positions (Mean Time: 13.1 s)

The robot approaches the components to perform the removal operation. First, the robot rapidly moves to a safety position displaced thirty centimetres in the Z-axis above the object and then moves to the desired location. After that, the robot moves back to the safety position. The timings for this operation have been divided into six sub-processes.

- Load the robot model (MoveIt!). Mean time: 3.2 s.
- Move to the safety position. Mean time: 1.8 s.
- Move to the component position. Mean time: 2.2 s.
- Removal operation. Mean time: not applicable, since it depends on the removal operations, which is not considered in this project.
- Load the robot model (MoveIt!). Mean time: 3.2 s.
- Move to the safety position. Mean time: 2.6 s.

### 3.3. Decision-Making: Optimal Path

The decision-making of the system (the order of the removal operations) affects directly onto the dismantling time. For this reason, the aim of this section is to analyse the order suggested by the task planner for the Audi A3 Sportback e-tron Hybrid Battery Pack.

### 3.3.1. Optimal Dismantling Plans

After manually dismantling and analysing the battery pack, the optimal dismantling plan is to first remove the screws; the second step is to remove the two connective components and the battery management system (the order of the removal operations for these three elements is not critical); and finally to remove the four battery modules in a arbitrary order.

### 3.3.2. Dismantling Plans Proposed by the System

A set of tests have been carried out under different conditions (i.e., different orientations, different ambient lights conditions, etc.), the system has given a good response.

It has been observed, within the proposed dismantling plans, that the system follows the guidelines defined in Section 3.3.1. Because the BMS and the two connective components (left and right) are not overlapping, the system is proposing two different plans that are equivalent. These are referred as the (A) and (B) plans and are illustrated in Figure 12.

In the (A) plan, the system begins removing screws. The screws are always the first components to be removed. Afterwards, the system removes one connective component (in some tests the left one in other tests the right one), the BMS, and the other connective component in that order. Finally, it removes the battery modules. See Figure 12. In the (B) plan, screws are still the first components to be removed. Then, the system proposes to remove the BMS and the connective components in that order. As in the previous plan, the battery modules are removed the last. Thus, the main difference observed between these two plans is that in the plan (B) the BMS is removed after the screws instead of connective plates. This has no impact on the final disassembly process.

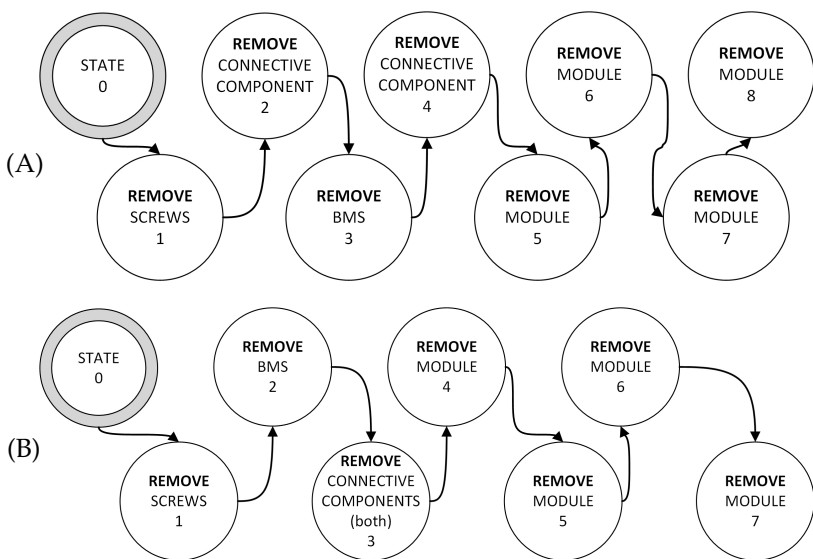

**Figure 12.** Dismantling plans (**A**) or (**B**) proposed by the task planner.

### 3.4. Accuracy

To analyse the system's accuracy, a 3D printed pointer has been used as Tool Center Point (TCP). The tool consists of a thin 25 cm long bar with a sharp end. With the tool mounted, the task planner has been run in debug mode. For safety reasons, the robot TCP has been moved 3 cm above in the Z-axis (word base frame) in order to avoid collisions with the battery pack in case of failure. Some of the tests are shown in Figure 13. In the

majority of the cases, the system has an accuracy of (<5 mm). The accuracy has been measured by the mean of a laser beam attached to the red bar and pointing towards the target position, whereas the distance of the laser pointer to the target position is measured with a caliper.

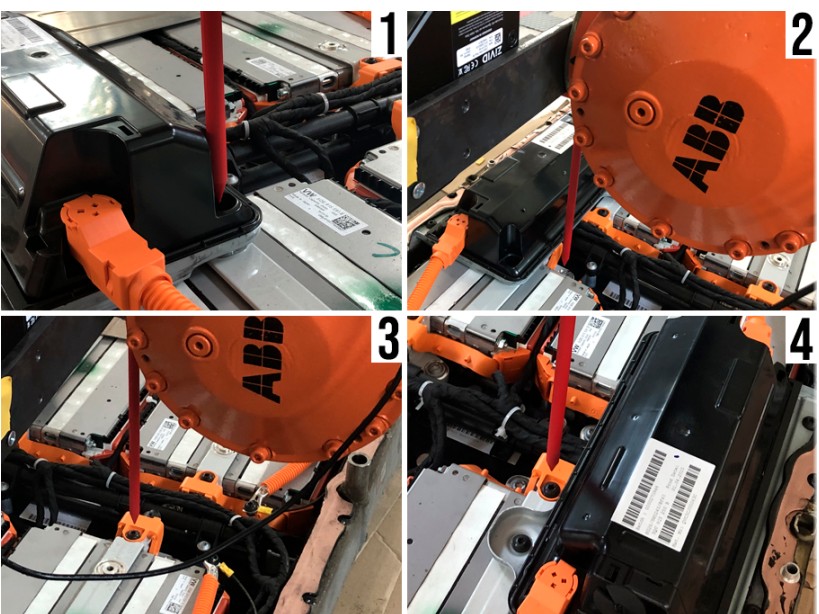

**Figure 13.** Accuracy tests showing the red pointer 3 cm above one detected screw. Four different views of the same position.

## 4. Discussion

In this paper the proposed objectives have been achieved. Different research areas in computer vision, robotics, circular economy and electr(on)ic components (battery) disassembly have been successfully unified.

The assigned main hardware elements such as an industrial robot, 3D camera and PC have been interconnected to carry out the principal system tasks, such as object detection, pose estimation, decision-making, and robot displacement. Therefore, the system is able to recognise the dismantling object main components, to find their position, and to move the robot to the defined positions in a specific order. Lab tests have been used to validate the designed task planner. In this case, experiments were limited to Audi LIB pack, however, a similar procedure might be applied to any EV battery pack. Compared to other disassembly processes as reviewed by Zhou et al. [11], the proposed task planner relies on state-of-the-art 3D camera system with high accuracy and does not require Computer Aided Design (CAD) models of the battery pack and its components. This presents a great advantage since EoL products are often different than their original CAD models, due to possible maintenance, deformations, or corrosion. Recognising the model and date of production of the EVB to be disassembled will help to determine a first disassembly sequence based on a self-updating database, but the system must also be flexible and robust enough to handle the above-mentioned variations or in the case of new or unrecognised model. Therefore, combined with reinforcement learning and machine reasoning algorithms the proposed disassembly framework will be able in future developments to learn how to disassemble new battery pack models, if not by itself, with only limited information from the human operator. The concept of cognitive robotics in disassembly has been developed and validated on End-of-Life treatment of LCD screen monitors [17]. However, some challenges remained as (1) the too high processing time making the process economically infeasible, (2) the remaining need for human assistance, and (3) the too high inaccuracy of the vision system leading to low success rate. This paper demonstrates that the recent advances in 3D vision system, fast object detection and localisation algorithms, as well as

task planner design place EVBs with inherent uncertainties and large variations in design as a good candidate for achieving eventually autonomous and complete disassembly through the cognitive robotic concept.

The proposed task planner for disassembly of EVB pack into modules can also be extended in future work to a deeper level of disassembly, i.e., to battery cell level or even to the cell components (cell casing, electrodes, electrolyte, separator) which will increase the concentration of active materials in the subsequent steps for battery recycling and hence reduce the complexity and energy consumption of the pyrometallurgical and hydrometallurgical processes. Removing manual operations in the pre-processing stages will move the optimum disassembly level determined in the article [3] deeper toward complete disassembly when still considering techno-economic and environmental constraints.

The algorithm You Only Look Once (YOLO) is implemented to detect and find the components placed in the dismantling arena. The results show that the algorithm performs well, giving expected results and detects main components. For example the presence of screws can be distinguished from the presence of screw holes where the screw has been removed. The developed vision system can hence also be used to validate removal operations. The information extracted from the object detection was used in a pose estimation to find coordinates of components, where 2D images and the YOLO results have been matched with the 3D data sets. In future versions of the task planner, object detection and pose estimation might be realised directly in 3 dimensions based on the point cloud data with techniques such as complex-Yolo [23] or DeepGCNs [24]. However, higher processing time or computing resources are to be expected.

When validating the task planner and measuring the timings, the robot model has been loaded every time that the robot had to move. Thus, in future stages the robot model should be loaded just once, at the beginning making the disassembly sequence at least 9.6 s faster. Moreover, the ROS main has been run in manual mode at 25% of the maximum speed of the robot. In automatic mode, i.e., at 100% speed, the total time of the disassembly sequence, excluding the removal operations time, is expected to decrease from 53.6 s to 34 s.

Using an eye-in-hand configuration performs well, and has some advantages (i.e., only one camera is needed), but it presents some drawbacks too. In an industrial application, the continuous moves and removal operations could have negative consequences like unexpected collisions of the camera with the environment or causing miscalibrations.

The efforts in future stages of the research should be focused on instrumentation and tool design for the dismantling system. It is also essential to detect flexible bodies such as high-voltage wires, and wires transmitting data. Thus, the direction of the research on the object detection and pose estimation part should concern how to find a feasible solution for such the objects and remove them.

## 5. Conclusions

A new task planner has been designed for the disassembly of electric vehicle Li-ion battery packs, with as main objective to increase the flexibility and robustness of the system. Lab tests have been used to validate the designed task planner based on a Audi A3 Sportback e-tron hybrid Li-ion battery pack. The results obtained in the tests demonstrate that the obtained solution is able to recognise which component to remove first and the complete disassembly plan without a priori knowledge of the disassembly strategy and battery CAD models. This method is therefore well suited for product with large variations and hence increases the disassemblability. The achieved performances measured in term of accuracy, time to generate the disassembly plan and success rate validated the task planner concept and its ability to make autonomous decision. Further testing on a larger set of EV battery packs with other geometries and connections and addition of learning capabilities will be needed to further increase the robustness of the proposed method and the technological readiness level. However, the results already cast a new light on the use of automation in the EV LIB batteries disassembly process by bringing the technology one step

closer to eventually fully automated operations and hence redefine the optimum level of disassembly for the batteries to enter the subsequent stages of recycling and metal recovery.

The experience in this field could also be adapted to be used for other dismantling processes and opens new doors and research challenges to other fields directly related to robotics.

**Author Contributions:** Conceptualization, E.M.B., M.C. and I.T.; methodology, E.M.B., M.C., and I.T.; software, E.M.B.; validation, E.M.B., M.C., and I.T.; formal analysis, M.C., I.T., and E.M.B.; resources, M.C. and I.T.; writing—original draft preparation, M.C., E.M.B., and I.T.; writing—review and editing, M.C., E.M.B., and I.T.; visualization, E.M.B., M.C. and I.T.; supervision, M.C., I.T.; project administration, E.M.B., M.C., and I.T.; funding acquisition, M.C. All authors have read and agreed to the published version of the manuscript.

**Funding:** This research was funded by The Research Council of Norway grant number 282328.

**Institutional Review Board Statement:** Not Applicable.

**Informed Consent Statement:** Not Applicable.

**Data Availability Statement:** Not Applicable.

**Acknowledgments:** Special thanks go to BatteriRetur AS for the battery pack used for experiments. This work contributes to the research performed at TRCM (Priority Research Center Mechatronics at the University of Agder, Norway).

**Conflicts of Interest:** The authors declare no conflict of interest.

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
