# Peer review of "Task Planner for Robotic Disassembly of Electric Vehicle Battery Pack"

_metals, doi:10.3390/met11030387_

Round 1

Reviewer 1 Report

The topic is relevant and there is no doubt that will be more than necessary in a near future.

I miss a more detailed description of the battery pack, specially if the battery pack is charged or discharged, if there is some damages or not...In other words, could the described task planner be performed in charge and/or damaged batteries? I can imagine a previous step of battery pack preparation needs to be added.

Are you planning to repeat the experiment with different battery packs with other geometries, other connections, etc?

Do the authors believe that a previous step of recognizing the kind of battery pack (thinking on industrial application) would be necessary? If so, maybe you may include your opinion on how the automated disassembly could be implemented on the industry.

Reviewer 2 Report

The paper is interesting and dealing with a serious and important topic however I would suggest to submit such a paper elsewhere than Metals. Topic seems to be out of the scope of the journal, therefore due to its excellent quality should be successfully addressed elsewhere.

Reviewer 3 Report

The authors present very interesting paper where they discuss on how robots could be used to disassemble EV battery packs. The work is very important, but the work is very short and lacks background. Also the results are presented clearly, but the discussion is not very deep. The work is not compared with other similar works. If the below mentioned things are taken into account, the paper will improve significantly and can be then publishes in this journal.

  1. The work only has less than 20 references, so the authors should discuss the background related to the paper. For instance now the papers for Lithium ion battery recycling is missing. As that is the application field, this should be included. There are several excellent reviews on that topic that should be mentioned, one example: http://doi.org/10.3390/batteries5040068
  2. As you are now going to dismantle a battery pack, it would be really nice to have a drawing on what parts there are and what are the different materials in those parts. As you are also submitting this to metals. What metals are you getting out in these stages.
  3. The real challenge and bottle next of the EV battery back recycling ist he recycle of the actual battery module (the one that has organic solvent electrolyte and nanosized active materials. Yes this is faster and cheaper than the manual dismantly, but the real challenge starts with the battery module. How to get nanosized Cobolt and Lithium recovered? Because THIS would really much be needed, as at the moment inteligence would be needed. Please discuss the applicability of this robote insistance system to LIb recycling.
  4. In your discussion session, you do not refer to any other studies, please compare your work to others. Is this already ready technology, or is further development needed, where do you see that it would be needed the most?
  5. In conclusions, the is only one chapter, please expand as this is very important. Please, do not review to other’s work for references (at conclusions).

Reviewer 4 Report

The subject of the article is very current, and the article itself deals with an interesting solution to the problem. The authors described the current state of knowledge well, and also justified taking up the research topic. The proposed research methodology does not raise any major reservations, and the description itself is sufficient to understand the idea of the solution. The research results are presented graphically, mainly in diagrams. The results obtained were discussed and briefly summarized. Some comments should be corrected:
- line 29, 34, 36 - it would be good to add article before [4], [6,7], [8], and similarly in the further part of the article
- fig. 3 requires additional description, it is incomplete in this embodiment
- fig. 9 - should be easier to read
- fig. 10 in the text appears after fig. 11, please replace
- fig. 13 - description of steps required

Round 2

Reviewer 2 Report

Excellent paper, accept as is

Reviewer 3 Report

thank you for addressing the comments.

This manuscript is a resubmission of an earlier submission. The following is a list of the peer review reports and author responses from that submission.